# Test cricketers score quickly during the 'nervous nineties': Evidence from a regression discontinuity design

**Leo Roberts** [1]*, **Daniel R. Little** [2], **Mervyn Jackson** [3], **Matthew J. Spittal** [1]

**1** Melbourne School of Population and Global Health, The University of Melbourne, Melbourne, Victoria, Australia, **2** Melbourne School of Psychological Sciences, The University of Melbourne, Melbourne, Victoria, Australia, **3** School of Health and Biomedical Sciences, RMIT University, Melbourne, Victoria, Australia

* leo.roberts@unimelb.edu.au

**Data Availability Statement:** The data underlying the results presented in the study are available from https://cricsheet.org/. To download all available test match data, navigate to https://cricsheet.org/matches/ and click on the "JSON"

## Abstract

The *'nervous nineties'* is a well-known cricket colloquialism that implies that batting within reach of 100 runs is mentally demanding. Despite common acceptance of this phenomenon, no study has used a historical test cricket dataset to examine how batting behaviour and performance changes on approach to a century. Accordingly, we explored opensource ball-by-ball data from 712 test cricket matches played between 2004 and 2022 to model the regression discontinuity of batting performance metrics either side of 100 runs. Models were fit using multi-level regression, adjusted for the clustering of balls within players (and where possible, the clustering of matches and innings within players). The analysis revealed that runs per ball and the probability of scoring a boundary increased as batters approached 100 runs. This was followed by a decline of -0.18 runs per ball (95% CI -0.22 to -0.14) and a three-percentage point decline (95% CI 2.2 to 3.8) in the probability of a boundary once a batter reached 100. The modelling revealed no evidence of a change in the probability of a dismissal before and after 100. Our results suggest many batters cope effectively with the psychological demands of playing through the nineties, including by batting aggressively and/or opportunistically to swiftly reach the milestone.

## Introduction

In sporting and other performances, being on the cusp of reaching a highly desirable goal can be psychologically challenging. It is well documented that even elite athletes sometimes lose mental control when close to meaningful success, then fail to execute their skills to a normal standard and miss the opportunity to achieve their goal [1]. Such occurrences are often labelled chokes or choking–stigmatising terminology that refers to the breakdown of normally expert skill under pressure to succeed [2]. While these events can take various forms, some of the most notorious examples in sport are characterised by the failure of an individual or a team to capture success that is virtually assured [3].

There are several reasons why putting the finishing touches on an important performance can be challenging. For one, the chance to achieve a personally meaningful goal can add

hyperlink located underneath the "Test matches" section. Code used to organise and analyse the data used in this study can be found on the Open Science Framework at DOI 10.17605/OSF.IO/7DXZN.

**Funding:** The author(s) received no specific funding for this work.

**Competing interests:** The authors have declared that no competing interests exist.

*psychological pressure* [4]–or an elevated weight on the need to perform well at a given moment [5]. Humans have imperfect thought control and are vulnerable to counter intentional thinking at inconvenient times [6]. As such, unhelpful thoughts can accompany the chance to realise an ambition (e.g., winning a status-building tournament), disturbing the high-performance mindset that brought the performer to the edge of success. The loss of mental control can manifest as attention shifting away from task relevant features and processes and onto the performance result and its consequences [7, 8]; harming performance if task relevant processing falls below critical levels [9]. Alternatively (or in addition), fixation on an outcome can encourage micromanaging thoughts related to how to execute the relevant motor skill [10]; an explicit focus thought to interfere with the smooth, effortless motions that experts are known for [11].

Another challenge of being in a great position to succeed is the perception that success should now be easily attained. This perception can exacerbate pressure because the expectation of success also comes with the possibility of an embarrassing (mental) failure that cannot be easily externalised. Consequently, there is a cost of performing poorly beyond just missing out on the valued reward. Moreover, the threat of being embarrassed is a deeply rooted source of *performance anxiety* [12], an issue (widely believed to accompany pressure) that activates kinematic, neurological and perceptual inefficiencies that challenge motor skill execution [respectively, 13–15]. As a result, athletes on the verge of achieving sought-after milestones may need to apply coping strategies to meet the elevated mental demands of their situation and maintain their performance enough to succeed [16].

With much interest in how athletes cope (or do not cope) in pressurised and or anxiety-provoking situations, numerous retrospective studies have assessed how athletes performed in (probable) clutch moments of competition [17–22]. Isolating high-pressure moments in sporting contests is valuable because it allows externally valid investigation of whether pressure affects performance and how athletes respond behaviourally (e.g., changing their pace of play or departing from typical behaviours). To date, retrospective studies have imagined high-pressure moments only as decisive parts of a close contest (i.e., in which neither side is particularly favoured to win), when the outcome is about to be decided or significantly influenced (e.g., a free throw late in a close basketball game, a penalty shoot-out in soccer or a break point in the deciding set of a tennis match) [17–22]. No retrospective study has, to date, specifically considered the impact of pressure created by being in a position of near certain success.

## The nervous nineties in cricket

In the sport of cricket, there is one passage of the game in which performers are always on the verge of reaching an important milestone: the 'nervous nineties'. This phenomenon, persistently referred to by cricket commentators [23] and acknowledged by some players [24, 25] references the mental challenge of batting when within ten runs of 100 (a century). Although cricket is a team sport in which 11 players attempt to collectively amass many runs, the individual accumulation of 100 runs is universally celebrated as a major batting achievement [25]. While the triumph essentially belongs to the individual, it can also benefit the team (beyond the helpful supplement of runs) by emotionally uplifting the batting side and signalling to the bowling team that the batters are in control [26].

Making centuries in cricket indicates prowess, effectiveness and mental toughness [27, 28], is an ingrained measure of batter status and legacy, and likely influences team selection. With great social value attached to making a century, it makes sense that a cricketer nearing 100 runs would experience more pressure than in some earlier parts of the innings when the milestone was not readily attainable (e.g., the 70s or 80s). Furthermore, while making centuries advances a batter's reputation, failing to capitalise on the opportunity is stigmatised. For

instance, negative press is given to batters who often reach 50 (a lesser milestone) without going on to reach 100 –known as a *failure to convert* [29, 30]–and to those who record numerous dismissals in the nineties [31, 32]. Hence, there is a reputational cost for failing to achieve a century when within (seemingly) easy reach that could be a source of performance anxiety. And yet, despite the psychological notoriety of batting in the nineties, there appears to be no psychologically oriented study of batting performance and behaviour during this part of an innings.

Since the nervous nineties has received minimal academic examination, it is unclear if or how batting performance or behaviour is impacted. It is posited here that if batters are generally influenced by pressure and/or associated anxiety when nearing 100 runs, a large historical dataset should reveal a global increase in the probability of dismissal, and/or a change in the run scoring rate, and/or a change in the probability of a boundary throughout the nineties. An increase in the overall probability of dismissal in the nineties is conceivable given several retrospective studies that have demonstrated the inferior performance of elite athletes under (assumed) pressure in tennis [18], soccer [21], basketball [22] and American football [19]. Furthermore, expert batting relies heavily on anticipatory eye movements since the ball moves faster than human eyes can track [33]. Considering that anxiety is known to reduce the efficiency of visual attention in various sports and is suspected to harm anticipatory abilities [34], it is plausible that cricketers approaching a century would be relatively prone to error.

An increased run-rate and/or probability of a boundary in the nineties is plausible considering numerous studies that indicate that athletes rush when impacted by pressure [8, 21, 35, 36], in order to escape the associated discomfort [21]. In the case of batting in the nineties, rushing could manifest as trying to score faster, potentially through aggressive shots (e.g., that result in boundaries) that allow rapid passage to the century. While rushing is a known response under pressure, slowing down is also a credible reaction. For example, some athletes deliberately slow down (physically and mentally) to directly offset anxiety-induced rushing [37]. In the case of batting in the nineties, slowing down could manifest as a period of patient, circumspect batting until the century is reached–which is the popular wisdom on how batters act during the nineties [38]. It is also possible that the run-rate or boundary making could slow due to a loss of motor proficiency under pressure. For instance, anxiety can result in inconsistent movement patterns in various motor skills [15, 39] as well as inefficient visual processing (see above)–either of which could interfere with batting timing and placement. In summary, an increase or a decrease in run-rate and the probability of a boundary are both plausible behavioural changes in the nineties.

## Previous studies of batting in the nineties

Two studies have explored batting statistics around landmark scores in cricket. Originally, Gauriot and Page [40], focused on batter incentives and strategy rather than performance with pressure, analysed run-rate and the probability of dismissal around 50 and 100 runs in international one-day cricket (a shorter version of the game than test cricket with 50 overs per side). Using ball-by-ball data from matches between 1971 and 2014, their results revealed that (a) batters were less likely to be dismissed just before 100 than after and (b) batters tended to score less quickly just before 100 than after. The interpretation was that batters play more conservatively (i.e., with less risk) when near 100 to protect their wicket, then, when over the threshold, resume a more aggressive (high risk) strategy with a greater chance of dismissal. While this is a useful analysis, there are good reasons to conduct a new study of batting performance/behaviour near 100 runs in test cricket. For one, test cricket is the most prestigious form played [41], hence making (or not making) centuries in test cricket should have especially strong

reputational consequences that heighten the pressure of approaching 100 runs. In addition, the one-day game has several constraints that likely influence batter behaviour and performance throughout the innings (e.g., batters reliably play under time pressure, bowlers are limited to 10 overs each, and fielding restriction rules impact the spread of the field during the innings). Essentially free from these constraints, distinct changes to batting behaviour and performance in test cricket (e.g., on approach to a century) are less likely to reflect structural features of game.

In the second study of batting around landmark scores, Stevenson [23] used predictive hazard functions to examine test match batting ability in 48 experienced, high-performing players throughout their various innings. This analysis, while based on innings totals, outputted an effective batting average (a proxy for batting ability) at different parts of the innings, with a dip in the 90s assumed to indicate nervousness. Stevenson concluded that there was little evidence to support a nervous nineties period (i.e., a drop in batting ability during the nineties) in those studied, even among cricketers well known for nineties dismissals. Notwithstanding this conclusion, the study used a select sample of batters and did not consider other metrics like run-rate. An opportunity remains for a broader analysis of batting when approaching a century that uses a large historical sample of test match batters and considers several metrics. Understanding the nature of batting in this psychologically loaded phase of the innings can (a) indicate how elite test cricketers are affected by proximity to a highly valued achievement; (b) establish if the nineties is a useful focal point for further investigations of performance under pressure in test cricket; and (c) potentially identify improved batting and bowling approaches during the nineties.

### The current study

The aim of this study was to examine if international test cricketers display a change in batting behaviour and performance when approaching 100 runs. To this end, a retrospective analysis of a large, contemporary ball-by-ball dataset was used to model batting metric fluctuations before and after 100 runs using a regression discontinuity design. The outcomes explored were the number of runs scored with each ball (a proxy for batting pace), the probability of a boundary (a proxy for batting aggression) and the probability of dismissal (a proxy for batting performance). The study was exploratory but noting evidence that elite performers in other sports tend to lose proficiency and behave differently in high pressure moments, it was anticipated that there would be a global increase in the probability of dismissal (performance) and changes (in either direction) in run-rate and the probability of a boundary (behaviours).

### Method

#### Dataset

Ball-by-ball data for all available test matches were obtained from the publicly accessible cricket data platform, cricsheet (www.cricsheet.org). The downloaded dataset contained player's names and associated batting statistics; information that is in the public domain and is freely accessible across numerous internet sources. At the time of the study, cricsheet held ball-level data for 91% of male test matches played between March 8, 2004 and March 24, 2022 and 80% of female test matches played between August 11, 2013 and Jan 27, 2022. In total, the dataset had ball-level data for 712 matches (704 male tests; eight female tests), 25,825 individual innings across 921 players, and 1,400,268 balls. Ball-level data were downloaded as JSON files and organised for analysis in R (version 4.1.3). Data verification was undertaken by generating a run total for each player in each innings in each match, then comparing these results to an alternative innings-level cricket database, accessed via the R cricketdata package [42]. Across

25,825 individual innings, a single run total discrepancy was identified (a difference of one run), emphasising the validity of the data. Three ball-level measures were constructed: the binary event of getting out or not, the number of runs scored from a given delivery and the binary event of scoring a boundary or not (i.e., $\geq$ four runs from a given delivery).

## Analysis

Regression discontinuity (RD) analysis was used to explore the change in batting performance and behaviour either side of a century. RD analysis is typically employed to assess the outcomes of people who are treated differently based on exceeding a certain numerical value (e.g., a cut-off assessment score that permits admission into a program). Imagining a clear relationship between a running variable (e.g., exam scores) and an outcome variable (e.g., future earnings), RD analysis explores the extent that there is a distinct break in the relationship of these variables at the cut-off score (i.e., a discontinuity of the regression fit). A key assumption of RD analysis is that participants have imprecise control over crossing the cut-off, and as such treatment assignment is effectively random in cases close to the edge [43]. Accordingly, any jump in the fitted regression at the cut-point can be assumed to reflect a causal effect of the treatment [44].

In the case of the exploring batting metrics around a century, reaching 100 runs results in a different treatment than falling just short (e.g., celebration, improved status/reputation, future team selection) and is a result imprecisely controlled by the batter. Therefore, if there is a break in the relationship between the number of runs scored (the running variable) and, for example, the probability of dismissal at the cut-point (100 runs), this would indicate an effect of the treatment (i.e., the consequences of scoring or failing to score a century) on batter performance.

In RD analysis, it is common and useful to examine not only a break in the average outcome at the cut-off, but also to consider the change in regression slope via an interaction term in the model [45]. The change in slope is relevant to the current study, given our interest in discovering varied patterns of batting behaviour/performance on either side of 100, where just less than 100 is presumably pressure-inducing while 100 or more is presumably more comfortable. The use of polynomial terms for non-linear relationships and random effects for clustered data is also recommended where appropriate to improve RD estimates [46].

Three models are reported that estimate the RD of the three outcomes. In line with typical RD models, these outcomes were examined over an interval near the cut-off, in this case 70 runs to 130 runs. To assess the stability of the model estimates, a sensitivity analysis was conducted in which the same models were re-run using an 80–120 run interval. All models included random intercepts to account for the multilevel structure of the ball-by-ball data (deliveries clustered within innings, matches, and players). Reflecting this structure, the *runs-scored model* was fit with a multilevel mixed-effects linear regression model with random intercepts for innings (1st or 2nd), match, and player. The *dismissal model* and *boundary model* were fit with multilevel mixed-effects logistic regression models with a player intercept only because of the problem of fitting multiple intercepts to binary data which often results in model non-convergence [47]. Initial models included terms for the score variable (transformed as a player's current score minus 100), the cut-off variable (0 for scores <100; 1 for $\geq$100) and their interaction. To account for a potential non-linear structure, a second set of models were run that also included a quadratic term for the score variable, along with the interaction between this variable and the cut-off variable. For each outcome, we compared this model to a simpler model that contained linear terms only. We selected this model if it provided a better fit to the data based on the AIC statistic. Models were fit with Stata-MP (Version 16.1).

## Results

### Sample characteristics

A total of 366 unique players (353 men; 13 women) scored between 70 and 130 runs at least once and were therefore included in the study. These scores were accumulated across 689 different matches (i.e., a match in which at least one person scored at least 70 runs) that comprised 2,719 innings and 144,641 balls. Players in this data sample scored an average of 112 runs (SD = 42 runs) per innings and 3.5 boundaries within the interval (SD = 3.0 boundaries). There were 1,767 dismissals of players who had scored between 70 and 130 (1,123 caught, 273 bowled, 240 leg before wicket, 47 caught and bowled, 43 stumped, 40 run out, one hit wicket). Of the remaining innings (952), 948 were not out when their innings ended or were dismissed with more than 130 runs. Four were retired hurt (classified as not out) between 70 and 130 runs. A total of 1,394 centuries were achieved (51%).

### Runs per ball

The results of our RD analysis showed an increasing number of runs per ball as the player approached 100 runs, a significant drop in the number of runs per ball after scoring a century, followed by some recovery in the number of runs per ball (**Fig 1**). Specifically, runs per ball increased non-linearly from a minimum of 0.59 (95% CI 0.58 to 0.61) at a score of 77 to a mean of 0.71 (95% CI 0.68 to 0.73) at a score of 99. At a score of 100 the mean runs per ball declined by -0.18 (95% CI -0.22 to -0.14) to 0.54 (95% CI 0.51 to 0.57). After this, the mean runs per ball gradually increased non-linearly, reaching a peak of 0.63 (95% CI 0.60 to 0.65) at a score of 125 runs. See the S1 Table for model coefficients.

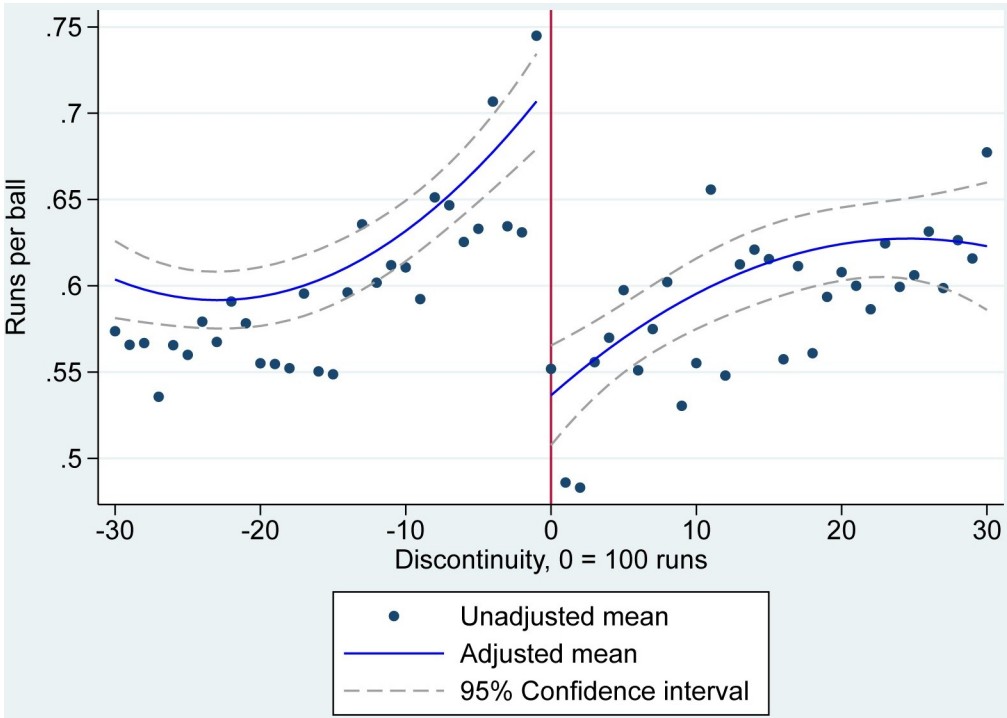

**Fig 1. RD estimates of the number of runs per ball.** Note: Unadjusted mean runs are averaged over innings, matches and players and do not account for each components' separate intercept. Adjusted mean runs are averaged over the model's random intercepts.

## Boundaries

Like the results for the runs per ball, the RD analysis showed an increasing likelihood of hitting a boundary as the player approached 100 runs, a significant drop in the probability of a boundary after scoring a century, followed by some return to the pre-century probability of a boundary (Fig 2). More precisely, the lowest probability of hitting a boundary was 6.4% (95% CI 6.1 to 6.7%), occurring when the player was on a score of 78 runs. This probability increased rapidly, reaching a maximum of 8.7% (95% CI 8.0 to 9.4%) at a score of 99. Upon reaching 100, the probability of hitting a boundary declined by three percentage points (95% CI 2.2 to 3.8%) to 5.7% (95% CI 5.1 to 6.2%). The probability of hitting a boundary then gradually increased again to seven percent (95% CI 6.6 to 7.4%) at a score of 123. See the S1 Table for the model coefficients.

## Dismissals

The overall pattern indicated no evidence that players had a different likelihood of dismissal after scoring 100. Rather, the pattern was indicative of a constant probability of dismissal throughout the period (Fig 3). Specifically, the probability of a dismissal at 70 runs was 1.3% (95% CI 1.2 to 1.5%) and remained at approximately this value until 99 runs. At 100 runs, there was no evidence of change, with the probability of a dismissal remaining at 1.3% (95% CI 1.1 to 1.5%). After 100, there was a non-significant increase in the probability of a dismissal, reaching a peak of 1.4% (95% CI 1.1 to 1.7%) at a score of 130.

## Sensitivity analyses

When the models were refit using an interval of 80–120 runs, instead of 70–130 runs, the results were essentially identical. See the S2 Table for a full set of results.

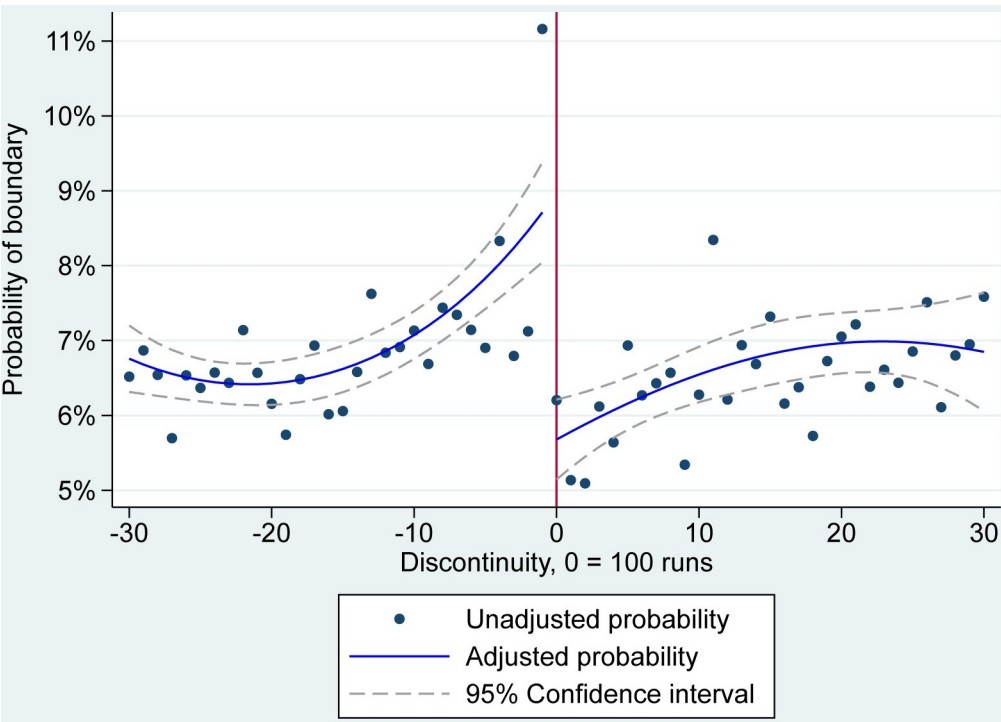

**Fig 2. RD estimates of the probability of a boundary.** Note: Unadjusted probabilities are averaged over innings, matches and players and do not account for each players' separate intercept. Adjusted probabilities of a boundary are averaged over the model's random intercepts.

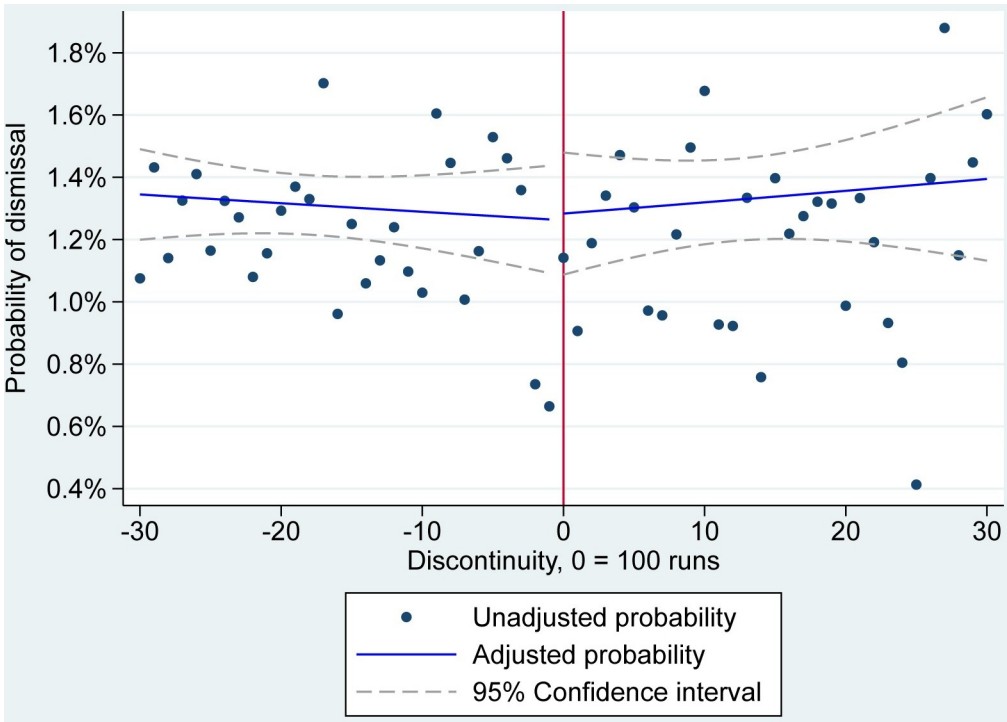

**Fig 3. RD estimates of the probability of dismissal.** Note: Unadjusted probabilities are averaged over innings, matches and players and do not account for each players' separate intercept. Adjusted probabilities of a dismissal are averaged over the model's random intercepts.

## Discussion

The aim of this study was to examine the behaviour and performance of international cricketers on the verge of a century with a regression discontinuity design. Behaviour and performance were measured using three variables: runs per ball (to assess batting pace), the probability of hitting a boundary (to assess batting aggression) and the probability of dismissal (to assess batting performance). The analysis showed that batters increased scoring and became more likely to hit a boundary on approach to 100, before reducing their productivity at 100 then gently recovering it. These results imply forceful batting in the nineties, followed by a more patient outlook upon reaching the century. The analysis also indicated that, in general, batters had a similar ability to retain their wicket before and after 100, suggesting widespread coping with any pressure experienced during the nineties.

### Increased productivity near 100

Accelerated scoring and an increased propensity to hit boundaries just before 100 suggests that batters introduce aggressive play to quickly reach their century (e.g., more forceful shots and/or harder running between wickets). This adjustment could reflect rushing, which has often been associated with choking in various sports [8, 48]. Rushing during a choking event is usually explained as playing or preparing to play quickly to escape the discomfort of anxiety [21]. This change in behaviour is thought to compromise task-relevant information processing (i.e., denying the brain access to data it could use to plan the motor execution) and is therefore considered an unhelpful avoidance strategy [35]. Translating to cricket, batters might force the pace of play to quickly escape the uneasiness of the nineties and secure the comfort of reaching

the century. This approach would be problematic if, for example, a batter attempts an aggressive shot to a delivery best handled defensively or attempts a run normally avoided (i.e., impatient batting).

While widespread rushing could explain the results, the analysis did not imply choking (most batters survived the nineties). Moreover, strategic changes made by the fielding team could facilitate more productive scoring near 100. In test cricket, it is common knowledge that the bowling side often attempts to shut down run-scoring as batters near 100 by bringing fielders closer to the crease [26]. The logic of this manoeuvre is presumably to prolong the batter's time in the nineties, build pressure and encourage a mistake. Ironically, a field tightly packed around the batter may create opportunities to quickly score runs (e.g., 'over the top') and depart the nineties sooner. Further research is needed to determine if batters tend to anxiously rush through the nineties, play opportunistically, or both.

Notably, Gauriot and Page [40] obtained a conflicting result in one-day cricket: batters reduced their run-rate just before 100 (protecting their wicket) then accelerated scoring upon reaching the milestone (risking their wicket). Strategic and structural differences between the two forms of cricket could explain the discrepancy. For instance, there is a long history of one-day batters manipulating the run-rate and choosing when to bat aggressively (e.g., in the last few overs) and when to bat patiently (e.g., in the middle of the innings). Several factors facilitate this. For example, bowlers in one-day cricket are restricted to 10 overs each (60 balls), which means that the best bowlers cannot always bowl (fatigue or injury are the only restrictions in test cricket). In collaboration with this, the field in one day cricket traditionally spreads out as the innings wears on (although various fielding restrictions have moderated the extent of this over the years). This configuration creates chances for an established batter (e.g., one approaching 100) to accumulate runs without playing aggressively or with much risk. Consequently, it is possible that one-day batters are better placed than test cricketers to manage the nineties as they would like (e.g., progressing slowly and safely to 100).

## Probability of dismissal unchanged

The fact that the probability of dismissal was unaffected in the nineties deviates from other retrospective studies of elite sport that found performance decline among elite performers in assumed high pressure moments [18, 19, 21, 22]. Nevertheless, batters' survival in the nineties, even if dealing with pressure and anxiety, is also not misplaced in the broader performance psychology literature, which presents a complex picture of the relationship between anxiety and performance. For example, it is widely accepted that performance is often maintained under anxiety given useful self-regulatory efforts from the performer [49, 50], and that individual factors will determine how much anxiety is felt [e.g., 51], and relatedly, how these symptoms are interpreted and responded to [50]. If batters do feel pressure and/or anxiety when close to 100, our analysis suggests that most cope effectively enough to retain their wicket. This explanation fits the widely held perception that cricketers find batting easier with time spent at the crease [52]. It may be that for the average international cricketer, the advantages of such acclimatisation overcome any anxiety-based processing inefficiencies. Furthermore, after a life of competition, elite cricketers (and other athletes) have typically developed a variety of strategies to manage performance in high-stakes moments [53].

## Strengths, limitations and future research

This study has several strengths. The use of the regression discontinuity design allowed us to precisely model changes in batter behaviour and performance at the century cut point. Unlike an earlier investigation of this topic in one day cricket, this study applied a multi-level

modelling strategy designed to account for the non-independence of observations, in this case, balls nested within innings, matches and players. The study also used a large, contemporary open-source dataset of 712 international test matches played between March 2004 and March 2022 and included every player that approached the century milestone in these games.

The study also has several limitations. First, like other retrospective investigations in the domain, this study assumed that the experience of pressure would be common at a high-stakes moment in the game. In our study, the assumption of pressure was based on the established reputational importance of making a century in elite cricket, anecdotal reports from past international cricketers that batting in the nineties can be psychologically demanding [24, 25], and the natural plausibility of pressure given the challenges associated with achieving highly valued success that is expected. Nevertheless, the perceptions of players were not assessed and remain unknown. Future qualitative research, in which elite cricketers are interviewed about their experiences batting during the nineties, could clarify the integrity of this assumption of pressure.

Second, as an initial exploration of the nature of test match batting during the nineties, the study took general approach, analysing changes in batting behaviour and performance before and after 100 runs at the population level. However, there may be a series sub-group effects that create a more complete picture of batting at this time. For example, it is likely that batters yet to score a century in their career will experience more pressure and anxiety than batters who have repeatedly surpassed the milestone. Likewise, batters who have been recently or often dismissed in the nineties are likely to face greater mental challenges than those with a track record of conversion. Future research could examine the nature of batting on approach to a century as a function of batter experience and/or ability.

Finally, the study only considered batting performance. As mentioned, it is possible that strategic changes from the bowling/fielding team contribute to the speeding-up effects observed in the study. It is also possible that the bowling/fielding team feels pressure to perform well as the batter approaches 100 runs, since the batter and/or batting team might receive a confidence boost from reaching the milestone. This pressure could manifest in poor deliveries, misfields, or dropped catches, any of which could result in an easier path to a century.

## Conclusion

Our analysis of batting metrics before and after 100 indicates that the century milestone (and its consequences) elicits a change in batting behaviour among international test cricketers. With 100 runs in reach, batters tend to accelerate scoring and boundary making, with no apparent cost to the probability of dismissal. Taking these results together, we surmise that the approach to 100 (including the nineties) is a productive, even successful time for international cricketers to bat. While it is possible that batters are rushing–and thus batting avoidantly and anxiously–they may also be exploiting newly formed gaps in the field introduced by the fielding side to subdue run-scoring. Either way, the results present an encouraging example of elevated performance when close to a highly valued achievement among an elite population.

## Supporting information

**S1 Table. Model coefficients for runs, boundaries and dismissals.** Note: Model for runs estimated using a multi-level mixed effects linear regression model. Models for boundaries and dismissals estimated using a multi-level mixed-effects logistic regression model.
(DOCX)

**S2 Table. Sensitivity analysis model coefficients for runs, boundaries and dismissals.** Note: Model for runs estimated using a multi-level mixed effects linear regression model. Models for boundaries and dismissals estimated using a multi-level mixed-effects logistic regression model.
(DOCX)

## Author Contributions

**Conceptualization:** Leo Roberts.

**Data curation:** Leo Roberts.

**Formal analysis:** Matthew J. Spittal.

**Methodology:** Leo Roberts, Daniel R. Little, Mervyn Jackson, Matthew J. Spittal.

**Writing – original draft:** Leo Roberts, Matthew J. Spittal.

**Writing – review & editing:** Daniel R. Little, Mervyn Jackson, Matthew J. Spittal.

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
