## [Decision Letter · Decision Letter 0]

16 Feb 2023

PONE-D-22-31412Test cricketers score quickly during the ‘nervous nineties’: Evidence from a regression discontinuity designPLOS ONE

Dear Dr. Roberts,

Thank you for submitting your manuscript to PLOS ONE. After careful consideration, we feel that it has merit but does not fully meet PLOS ONE’s publication criteria as it currently stands. Therefore, we invite you to submit a revised version of the manuscript that addresses the points raised during the review process.

We look forward to receiving your revised manuscript.

Kind regards,

Jonathan Douglas Connor, Ph.D

Academic Editor

PLOS ONE

Journal Requirements:

2. In the ethics statement in the manuscript and in the online submission form, please provide additional information about the data records used in your retrospective study. Specifically, please ensure that you have discussed whether all data were fully anonymized before you accessed them.

Additional Editor Comments (if provided):

Thank you for your submission. Two reviewers have returned their assessment and provided some feedback on how to improve the manuscript. I invite you to address these reviewer comments and make any changes you believe appropriate to your manuscript.

Reviewers' comments:

Reviewer's Responses to Questions

**Comments to the Author**

1. Is the manuscript technically sound, and do the data support the conclusions?

Reviewer #1: Yes

Reviewer #2: Yes

2. Has the statistical analysis been performed appropriately and rigorously? 

Reviewer #1: Yes

Reviewer #2: Yes

3. Have the authors made all data underlying the findings in their manuscript fully available?

Reviewer #1: Yes

Reviewer #2: Yes

4. Is the manuscript presented in an intelligible fashion and written in standard English?

Reviewer #1: Yes

Reviewer #2: Yes

5. Review Comments to the Author

Reviewer #1: The sample size used for the analysis is extensive and provides noteworthy rigor to the analysis, although some minor questions over the inclusion of female data in the analysis and further explanation required on the analysis period chosen.

I am aware of the statistical method used and it seems to me the appropriate analysis has been applied. I am no expert in this particular method though and may result in me overlooking the finer details.

This manuscripts well written.

Reviewer #2: - The introduction is quite clear and well-described.

- Data, please explain how authors ensured the data validity, just downloading the data from the internet?

- Does the RD model account/ control for the year-by-year performance? As a temporal series autocorrelation, regression to the mean and outliers should be included to improve the robustness of the model. Please clarify these issues.

- Why conduct an RD with repeated measures? Probably a multilevel model with random factors and clustered variables would fit either for these repeated measures from the data. Please justify why this model was used and not the MLM, random forest, or neural network...

- L235, please write 1,123

- L238, please write 1,394

- Results and discussion, my main concern about both sections is how the evolution of years (performance) is related to the model and presented to justify how it affects along the years where training methods, physical and technical training, as well as psychological training have evolved and can be affecting the identified trends.

- Conclusions, L393-396, please avoid conclusions that were not obtained from your data. This suggestion can be included in the discussion but not in the conclusion.

6. PLOS authors have the option to publish the peer review history of their article (what does this mean?). If published, this will include your full peer review and any attached files.

Reviewer #1: No

Reviewer #2: No

---

## [Author Response · Author response to Decision Letter 0]

4 Mar 2023

We thank both reviewers for taking the time to read and consider our work and offering their constructive suggestions. From these comments, we have made several adjustments to the paper that we believe will improve its broader acceptance and appeal. These changes include providing more detail about the ball-by-ball dataset used, reporting a validation exercise that we performed to verify the data used, enhancing the argument used to justify an examination of this topic in test cricket, and the removal of some of the more speculative sentences. Below we provide specific responses to each reviewer comment with some more detailed answers to the questions raised by reviewer 2 in relation to our chosen analyses. 

Reviewer 1:

Is there a reason for choosing this time period? Maybe it is obvious in that this is all the data that was available in ball-by-ball format, if this is the case it might be good to tell the reader this.

More information has been added about the ball-by-ball data available, including that we used everything that was accessible at the time.

Is there a reason for including male and female data together? Would it not be worth examining the female data separately (or excluded) given a century is a much rarer event in this game? Also given the disproportionate number of female games compared to male games any effect in the female game would be washed out and the results likely the same as if you didn’t include them.

We did out of curiosity try the analysis using men-only, and as suggested, this had no impact on the findings. Given this, we left the relatively small sample of women in the dataset because we couldn’t see a good reason for publishing a male-only analysis. Likewise, a female-only analysis wasn’t feasible with the numbers available.

It is mentioned that previous work has analysed scoring in the nineties in ODI cricket, but test cricket might provide a better context to examine how batting is impacted by pressure due to its prestige. I question whether it is “better”, a century is extremely important to a batter regardless of the format. In fact, you could say it is easier to score a century in test cricket as the game dictates you don’t have to score as quickly. I would suggest restructuring this paragraph (line 141-146) to reference the task related constraints that dictate the game be played differently to ODI cricket.

As suggested, we have rewritten this section to suggest that test cricket is a good context to explore the impact of a century on batting and performance – including that it is free of some constraints of 50 over cricket. There is now less insinuation that it is necessarily the best context.

Line 71-75: Mentions numerous sport specific areas of retrospective studies that have analysed high-pressure moments, it might be good to reference these in this sentence.

These references have been added.

Line 316: “Sorting out” seems a little colloquial…rephrase.

We have rephrased.

Line 325-328: It is mentioned that less potent bowlers present the opportunities for batters reaching 100 runs to bat more conservatively in ODI cricket. This is quite speculative, and I think a good argument could also be made for the opposite occurring. 

We have removed the sentence about the ability to undertake low risk batting when less potent bowlers are operating. 

Line 332-333: In the below sentence, consider changing “are” to “it is possible that one-day batters…”

“Consequently, one-day batters are better placed than test cricketers to manage the nineties as they would like (e.g., safely progressing to 100)”

We have adopted this suggested wording.

Reviewer 2:

Data, please explain how authors ensured the data validity, just downloading the data from the internet?

We have added information about a data validity exercise we undertook. While the data we used was the only ball-by-ball dataset we could find, we were able to completely verify the data at the innings level (i.e., by comparing individual innings run totals in the dataset we used with an alternative cricket database). 

Does the RD model account/ control for the year-by-year performance? As a temporal series autocorrelation, regression to the mean and outliers should be included to improve the robustness of the model. Please clarify these issues.

The results presented in the manuscript do not control for time effects. However, to clarify this issue, we have re-run all our models controlling for time. Time trends were not significant for two of the outcomes (runs per over, p = 0.405 and probability of being dismissed, p = 0.620). Time was significant for the probability of getting a boundary (p = 0.017), and the direction of the relationship suggested that the odds of getting a boundary had declined very slightly per year (OR = 0.993). Nonetheless, when we looked at the values for the other coefficients, none had changed between models that adjusted for year and those that didn’t. Our interpretation is that time is not a confounder in this case. That makes sense. It is difficult to see what temporal changes may have changed the association at the discontinuity, which we note, is different to a more general question about how scoring rates and risk of dismissals have changed over time. Issues like regression to the mean are interesting but not directly relevant here as (a) we use a subset of all test data – the data of those who scored between 70 and 130 runs and (b) all models are fit as multilevel models with runs nested innings, innings nested within matches, and matches nested within players. Finally, we are of the school of thought that outliers should not be removed as this has the potential to introduce bias into the study, and the data are what they are.

Why conduct an RD with repeated measures? Probably a multilevel model with random factors and clustered variables would fit either for these repeated measures from the data. Please justify why this model was used and not the MLM, random forest, or neural network...

In fact, we did fit multilevel models with random factors/clustered variables, with RD assessed using (a) a term that captured whether or not the century had been reached (<100 = 0; >= 100 = 1) and (b) the interaction of the batter’s proximity to a century and the binary century variable. In effect, we ran a multilevel regression discontinuity model. Where useful, quadratic terms were also included. This method with justification is explained in the Analysis section.

- L235, please write 1,123

- L238, please write 1,394

These have been changed.

- Results and discussion, my main concern about both sections is how the evolution of years (performance) is related to the model and presented to justify how it affects along the years where training methods, physical and technical training, as well as psychological training have evolved and can be affecting the identified trends.

We have covered this point above.

- Conclusions, L393-396, please avoid conclusions that were not obtained from your data. This suggestion can be included in the discussion but not in the conclusion.

This speculative sentence has been deleted.

---

## [Decision Letter · Decision Letter 1]

12 Jun 2023

Test cricketers score quickly during the ‘nervous nineties’: Evidence from a regression discontinuity design

PONE-D-22-31412R1

Dear Dr. Roberts,

We’re pleased to inform you that your manuscript has been judged scientifically suitable for publication and will be formally accepted for publication once it meets all outstanding technical requirements.

Kind regards,

W. David Allen

Academic Editor

*PLOS ONE*

Additional Editor Comments (optional):

Reviewers' comments:

Reviewer's Responses to Questions

**Comments to the Author**

1. If the authors have adequately addressed your comments raised in a previous round of review and you feel that this manuscript is now acceptable for publication, you may indicate that here to bypass the “Comments to the Author” section, enter your conflict of interest statement in the “Confidential to Editor” section, and submit your "Accept" recommendation.

Reviewer #2: All comments have been addressed

2. Is the manuscript technically sound, and do the data support the conclusions?

Reviewer #2: Yes

3. Has the statistical analysis been performed appropriately and rigorously? 

Reviewer #2: Yes

4. Have the authors made all data underlying the findings in their manuscript fully available?

Reviewer #2: Yes

5. Is the manuscript presented in an intelligible fashion and written in standard English?

Reviewer #2: Yes

6. Review Comments to the Author

Reviewer #2: Thanks for your revised version of the article and the justification of all reviewers' comments. Please incorporate into the analysis and results the values controlling for temporal series, this brief information is required for readers (or just add it as a supplementary table).

7. PLOS authors have the option to publish the peer review history of their article (what does this mean?). If published, this will include your full peer review and any attached files.

Reviewer #2: No

---

## [Editor Report · Acceptance letter]

16 Jun 2023

PONE-D-22-31412R1 

Test cricketers score quickly during the ‘nervous nineties’: Evidence from a regression discontinuity design 

Dear Dr. Roberts:

I'm pleased to inform you that your manuscript has been deemed suitable for publication in PLOS ONE. Congratulations! Your manuscript is now with our production department. 

Kind regards, 

on behalf of

Dr. W. David Allen 

Academic Editor

PLOS ONE